# RivEr/Generation_LAB-Linking Resilience with Inclusiveness in the Urban-Built Environment of Rome

**Carmela Mariano *** and **Francesca Rossi**

Department of Planning, Design, Architecture Technology, Sapienza University of Rome, 00185 Roma, Italy
* Correspondence: carmela.mariano@uniroma1.it

**Abstract:** The impact of metropolization processes and climate change effects on natural and anthropic environments, together with energy waste, the excessive consumption of agricultural and natural soils and their progressive waterproofing and a reduction in vegetation cover, highlights the need for sustainable management of existing resources, in terms of equitable and ethical development, towards sustainable and inclusive communities able to adapt to the negative effects of emergency phenomena. This contribution presents the results of the activities conducted in the RivEr/Generation_LAB, a project organized by three CIVIS members (Sapienza University of Rome, Universitè libre de Bruxelles, Universidad Autonoma de Madrid) as a part of the CIVIS Project "RivEr/Generation_LAB. Linking resilience with inclusiveness in the urban built environment of Rome, Brussels, and Madrid", financed by the Hub4 Cities, Territories & Mobilities' Call for proposals 2021. The project proposes a methodology of intervention in the Flaminio district, in particular in the Olympic Village and its relationship with the Tiber River, in line with the Sustainable Development Goals, the Millennium Ecosystem Assessment and the New European Bauhaus, to establish new relationships between cities and the natural environment, favoring sustainable and inclusive public spaces.

**Keywords:** resilience; climate change; vulnerability; water management and urban planning; environmental technological design; nature-based solutions

## 1. Introduction

In the last decades, the urban climate was a deeply studied issue, especially regarding air temperature variations from urban to rural areas, defined respectively as the urban heat/cool island effect [1], which exposes European cities to increasing climate risks [2].

Legambiente [3] put into evidence the critical issues linked to climate discomfort in Rome's urban open spaces. The record temperatures values of summer 2003 severely affected the population in terms of mortality, with 944 excess deaths observed (+19%).

The thousands of city microclimates, which vary according to its urban structure and morphological characteristics, are the expression of a non-uniform equilibrium that ought to be developed and optimized, as it is interrelated to the whole urban area. Green and blue infrastructures—especially waterscapes, trees and permeable soils—have proved to have a positive effect on improving external temperature conditions [4], absorbing sun rays hitting urban surfaces and tempering negative thermal conditions through radiative cooling. The need to re-design cities' outdoor spaces, in order to mitigate global warming, improving resilience and residents' well-being [5], is the key objective of a radical change in city planning and design, which is necessary to effectively address the above-mentioned issues, promoting innovative climate-adaptation strategies, and fostering higher levels of urban welfare [6,7].

In particular, the awareness that a sustainable management of waters is needed to enhance rivers' contribution to contrast climate change for their high biodiversity and strong productivity of ecosystem services is motivated by the growing incidence, especially in urban contexts, of floodings associated with resource depletion [8,9].

Rivers, in fact, as historical structural elements of urban growth, closely link the urban morphology to the ecological value of its natural environments, and this is why in 2007 the European Parliament called on Member States to foster a long-term planning approach to water management in order to limit the severe impacts of flooding on human well-being, the environment, heritage and socio-economic activities, by integrating all cognitive data on hazards, vulnerability and hydraulic risks [10,11]. The key aim of safeguarding the provision of high-quality water is pursued by coordinating water management policies with ecosystem-based approaches, such as developing renewable technologies, providing sustainable transport and promoting nature-based solutions in urban planning [12,13].

The restoration of natural ecosystems, related to a more sustainable use of resources and to a reduced fragmentation of green networks, represents a pivotal action towards the construction of regenerative urban contexts, enabling local communities to adapt to climate change, through new forms of economic, cultural and social resilience [14,15]. The developing identity of cities and communities, historically recognizable along their riverbanks for their significant heritage and cultural value, can work as a driver to counter the growing processes of the degradation and vulnerability of urban contexts [16–18].

To this end, the promotion of an integrated strategy to protect the natural and urban environment through the restoration of water resources has been unanimously expressed by the Sustainable Development Goals of the 2030 Agenda of the United Nations, the New Urban Agenda, the European Green Deal and, more specifically, by the Water Framework Directive and the Biodiversity Strategy for 2030 [19–22].

In this framework, the implementation of innovative research and didactic paths, aimed at the protection and restoration of river landscapes, the correct management of water resources and the valorization of urban territories, together with safeguarding actions, can contribute to the development of a new awareness of citizens to the ecological, morphological, historical, cultural and social role of riverfronts in producing a new sense of belonging and care. A planning approach that links a responsible, inclusive and sustainable strategy for the construction and networking of new resilient urban spaces through green infrastructures, riverfronts, new green spaces and cycle paths, represents a resilient response to cities' regeneration [23–25].

In particular, this contribution presents the results of the education, research and experimentation activities conducted in the RivEr/Generation_LAB, a project organized by three CIVIS members (Sapienza University of Rome, Universitè libre de Bruxelles, Universidad Autonoma de Madrid) as a part of the CIVIS Project "RivEr/Generation_LAB. Linking resilience with inclusiveness in the urban-built environment of Rome, Brussels and Madrid", financed by the Hub4 Cities, Territories & Mobilities' Call for proposals 2021 [26].

CIVIS is a European Civic University co-funded by the Erasmus+ Programme of the European Union and formed by the alliance of 11 leading research higher education institutions across Europe: Aix-Marseille Université, National and Kapodistrian University of Athens, University of Bucharest, Université libre de Bruxelles, Universidad Autónoma de Madrid, Sapienza University of Rome, Stockholm University, Eberhard Karls Universität Tübingen, University of Glasgow, Paris Lodron University of Salzburg and the University of Lausanne. Therefore, CIVIS' main objectives are as follows: to create a truly unique European inter-university campus where students, academics, researchers and staff will move and collaborate as freely as within their institution of origin; to develop a deep level of European integration, involving joint learning pathways; to promote complementary research facilities and diverse degree pathways.

*General Goals of the RivEr/Generation_LAB*

The project RivEr/Generation_LAB, through the objectives and values of the CIVIS Programme, acts in sharing, among different universities, to foster an integrated and holistic approach to achieve unity by sharing and putting together new different methodologies and backgrounds belonging to and from different universities joining a specific CIVIS agreement. In this case, the integrated approach proposed by the Sapienza University

of Roma (SUR), as the hosting University, and the two partners, the Universitè Libre de Bruxelles (ULB) and the Universidad Autonoma de Madrid (UAM), worked together on a transdisciplinary methodology of intervention, which combines the disciplinary approaches of the three partners in the fields of urban planning and environmental design, ecology and landscape architecture, mobility and the urban economy, in line with the Sustainable Development Goals, the Millennium Ecosystem Assessment [27] and the New European Bauhaus [28], to establish new relationships between cities and the natural environment, favoring sustainable and inclusive public spaces (Figure 1).

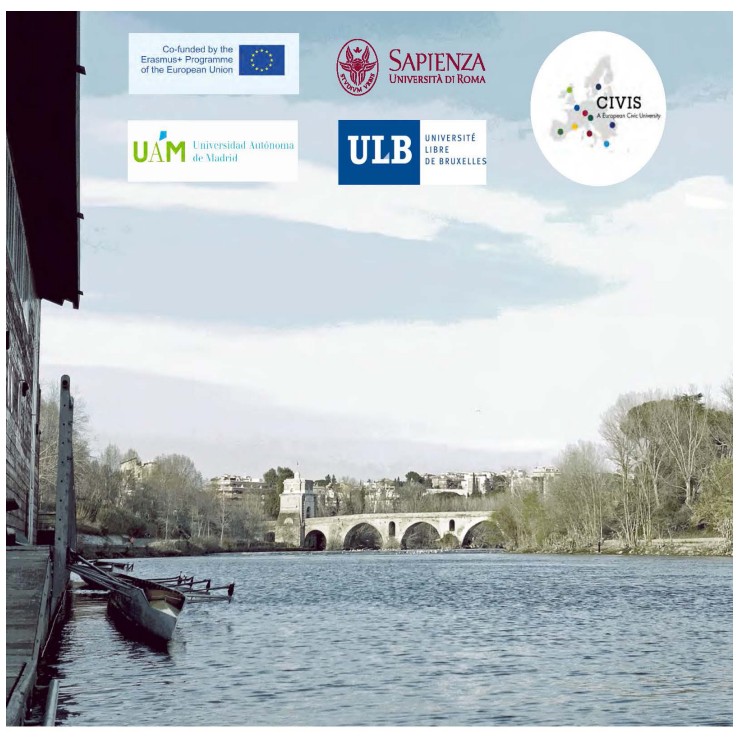

**Figure 1.** The Workshop Poster with the Initiative Partners. Source: authors' elaboration.

It is an approach that aims to build "bridges of knowledge and action" between the seemingly competing systems of urban planning and environmental design; ecology and landscape architecture; mobility; and the urban economy, which interact and influence each other in our everyday urban and peri-urban environments.

In particular, RivEr/Generation_LAB brings together three river areas in Rome, Brussels and Madrid to address, through cooperation in the field of education, the current challenges of urban sustainability by implementing a problem-solving approach that integrates systems thinking with real-life scenarios and experimental field activities.

The project proposes a comprehensive and inclusive approach to promote diversity in education by bringing together different disciplines and perspectives into a single analytical framework, which explores the commonalities of approaches to the regeneration of urban riverscapes across different dimensions and critically examines the implications of the interrelation between different sectors and/or domains. The development of original case studies and the critical analysis of urban regeneration policies relate to the existing dynamics at the local level, specific to each of the three river landscapes.

Through the proposed multidisciplinary comparative approach, participants can understand the values of becoming active citizens in local, national and global communities. The comparison of different urban and landscape contexts, starting from the common goal

of riverfront regeneration, fosters awareness and understanding of the qualitative aspects required in such ambitious projects, emphasizing the multiple perspectives and values involved in climate-friendly holistic urban transformation processes.

Thus, the project aims to foster civic engagement initiatives involving students as part of broader regeneration activities, sharing knowledge, skills and motivation necessary for responsible engagement in diverse societies.

In summary, RivEr/Generation_LAB pursues the following objectives:

- To promote key competencies among students and faculty in the field of sustainable urban development.
- To build capacity among academics, enabling them to successfully implement interdisciplinary and transformative learning contexts.
- To create and share knowledge of teaching and learning, documenting CIVIS cooperation as a replicable educational resource.
- To include different disciplines and/or fields.
- To share a comparative approach.
- To promote Civic engagement.

The short-term results are as follows:

- Efficient dynamic education for the regenerative development of urban rivers.
- Development of multidisciplinary and innovative skills for staff and students.
- Opportunity, for students and teachers, to master integrative methods and theories not usually included in their curricula (inter alia, drawing on knowledge and value systems from other disciplines, systemic thinking, scenario building and visioning, as well as experimenting with integrated social/environmental/economic impact assessments).

The long-term results are as follows:

- Promotion of multidisciplinary expertise through the CIVIS cooperation.
- Development of best practices in spatial design and planning, characterized by integrated strategies and nature-based solutions.
- Ecological and social reconnection of urban riversides.

## 2. Materials and Method

### 2.1. Materials

The methodological framework of RivEr/Generation_LAB embraces a pedagogical and training cycle, which allows the CIVIS cooperation values to be tested through an innovative combination of the virtual and real mobility of students and teachers.

The virtual element consists of six preparatory webinars organized in the virtual classroom by the three partners and related to the common field of urban regeneration (for more details see Appendix A). These online elements provide the theoretical framework and operational guidelines for experimenting with an urban regeneration strategy conducted in an on-site design workshop in Rome [29], coordinated by the PDTA of Sapienza University of Rome (SUR), with the collaboration of lecturers from the Université libre de Bruxelles (ULB) and the Universidad Autónoma de Madrid (UAM), the managerial and organizational support of tutors belonging to the PDTA and the active participation of the main actors and recipients of this activity, the students from each university participating in the project.

In particular, the webinars provided the basic knowledge and skills related to the three subject areas of urban planning and environmental design: ecology and landscape architecture; mobility and urban economy for landscape; and environmental regeneration and enhancement of urban contexts to promote the economic, cultural and social resilience of river landscapes.

With reference to the subject area of urban planning and environmental design, coordinated by SUR, the following topics were addressed:

- The framework of the main historical planning tools and the urban transformation of the Flaminio district in Rome, with particular reference to the Olympic Village;

- Urban regeneration strategies such as defense, adaptation and relocation as a response to climate change phenomena;
- Design criteria in flood risk areas through the analysis of national and international best practices;
- Operational references for protecting and restoring water-related ecosystems affected by climate change, consistent with the targets of Goal 6 of the UN 2030 Agenda;
- Case studies of adaptation solutions in urban contexts and rural areas using water as a strategic factor for urban regeneration.

With reference to the subject area of ecology and landscape architecture, coordinated by the ULB, the following topics were addressed:

- Nature-Based Solution [30] for the regeneration of aquatic ecosystems and water resources through the management of precipitation, humidity, water storage, infiltration and transmission, so that improvements are made in the location, as well as the timing and quantity of water available for human needs;
- Water management in the Brussels case study through the engineering approach and the architectural-landscape approach to the design of public spaces;
- Best practices of living with water, the case study of Venice.

With reference to the subject area of mobility and the urban economy, coordinated by UMA, the following topics were addressed:

- The topic of mobility with a multidisciplinary approach from social, societal and economic perspectives;
- Sustainable mobility systems in the smart cities model (Smart Mobility Plans and Mobility as a Service);
- Best practices of sustainable mobility, the case study of "Madrid 360 Sustainable Mobility: Cleaner air for all";
- The topic of "Urban Economics" related to the "Madrid Rìo" project applied to the case study Manzanares River in Madrid.

In addition, to make the students' design work easier, tutors and teachers have preliminarily prepared information/didactic materials to support the project activities.

These materials consist of the following:

- Town planning documentation relating to the project area | General Regulatory Plan (PRG) [31] (Municipality of Rome, 2008); Rome Strategic Plan 2010–2020 [32] (Municipality of Rome, 2010), Flaminio Foro Italico Urban Project—(PUF) [33] (Municipality of Rome, 2014);
- Cartographic documentation related to the project area | Orthophoto, the DWG file of the project area, PDF files of the project area on a scale of 1:10.000, 1:5.000 and 1:2.000, the 3D file of the project area and poster and presentation layouts for the final presentation;
- In-depth report on the microclimatic analysis preliminarily carried out on the project area;
- Webinar recordings.

### 2.2. Method Workshop Phases

2.2.1. Specific Goals of the Workshop "RivEr/Generation_LAB"

- Interdisciplinarity, related to the cooperation values of CIVIS, as evidenced by the topics addressed during the six webinars ranging from Urban Planning for River/Generation, Mobility, Urban Economy, Ecology, Landscape Architecture and Environmental Design. The integration, especially of economic disciplines, brought an innovative element from the point of view of sharing knowledge between actors with different backgrounds.
- Definition of a model of actions, costs and benefits (Toolkit of Actions), which recalls the concept of a circular economy, a model that implies sharing, borrowing, reusing, repairing and reconditioning existing places as long as possible. These principles, intended in terms of public space planning, were found in the reuse of public spaces and in the redefinition of the uses of places.

- Providing knowledge of a replicable methodology that enriches the background of each participating student.
- Sharing values and methods. The division into groups was motivated by the desire to make effective the sharing of different methods and backgrounds but, above all, the sharing of disciplinary knowledge. To this end, the groups were divided heterogeneously, ensuring in each group the presence of students with a background and course of study mainly in the disciplines of economics, spatial planning and architectural technology. This enabled the creation of discussion and knowledge exchange tables, to formulate constructive considerations capable of developing inclusive and integrated group work.

The methodological structure adopted during the design workshop aims to acquire a deep knowledge of the urban context, recognizing its structural and morphological components.

Indeed, thanks to a preliminary cognitive analysis of the urban context and a consequent critical evaluation of its components, it is possible to clarify coherent goals, based on which a project proposal that fits perfectly with the surrounding area can be elaborated, which provides a functional mix, which integrates private and public spaces, which contemplates a hierarchical system of mobility and accessibility, and which guarantees the integration of green areas with the settlement system.

What is expressed is reflected by a system and components' approach divided into 3 main phases:

- Phase 1 | Analysis and critical evaluation of the territory
- Phase 2 | Planning
- Phase 3 | Project

The workshop started with the presentation of the project area, the Olympic Village, located in the Flaminio district of Rome, and the identification of the main territorial and morphological characteristics of the area, providing a historical and urban contextualization. After the on-site visit, tutors showed the report of the microclimatic analyses carried out in the Olympic Village, explaining the results of the following analyses: sun path diagrams and shadow range (using software Ecotect) and the microclimatic parameters, such as air temperature, wind speed, wind temperature and relative humidity (using the software ENVI-met), to provide an evaluation framework of the outdoor thermal comfort

Afterwards, tutors explained the methodology and the design process adopted in the workshop, also showing students some examples, and at the end of the workshop, on July 2022, it was the turn of each group of students to present the projects to teachers, tutors and to the other students.

This report's "Results" section describes the results in detail.

During all phases ("Analysis and Critical Evaluation", "Planning" and "Project"), teachers and tutors carried out support and monitoring activities.

In this regard, each day of the workshop was characterized by the following support and monitoring phases:

- Description of day's activities | At the beginning of each day teachers and tutors explained the activities scheduled for that day, goals to reach and the timing of working activities;
- Support and monitoring during activities | During each day, teachers and tutors carried out support and monitoring activities for each group aimed at achieving the daily goals defined and discussed with the students at the beginning of each day;
- Work check and Presentation | Each day of the workshop ended with a check of the progress of each group by teachers and tutors, a collective discussion and the presentation of the work to the other students, teachers and tutors to encourage comparison, exchange and brainstorming.

Concerning the methodological structure adopted, a detailed description of the three phases of "Analysis and critical evaluation of the territory", "Planning" and "Project" is given below, explaining the timing and the purposes of each.

2.2.2. Phase 1 | Analysis and Critical Evaluation of the Territory

The first phase was characterized by two consequential steps:

1. In-depth analysis of the urban context- | This step provides an elaboration of three separate analyses for the "Environmental System", "Settlement Systems" and "Infrastructures, and Public services System", aimed at bringing out the components of each system that have a structuring role for the urban definition.

This type of analysis is very thorough and takes a long time, so the students were provided with supporting urban planning material, which already highlighted the main components of each system; in this way, each group was able to study and summarize the material.

2. Critical evaluation of the urban context | In this step a critical evaluation of the systemic components analyzed in the previous step of "In-depth analysis of the urban context" was carried out. This step is aimed at highlighting for each system ("Environmental System", "Settlement System", "Infrastructures and Public services System") existing values/qualities on which to base the redevelopment "project" (*resources*); the recognizable potentialities in some components and/or aggregations of components (according to the relationships between them) that may represent significant opportunities to be "exploited" in the "project" (*potentials*); the risks and negative situations recognized in certain components and/or portions of the territory that the "project" must correct, orientate and/or eliminate (*criticalities*).

2.2.3. Phase 2 | Planning

In the second phase of "Planning", project strategies and goals were defined, again at a systemic level, which led to the drafting of a "Preliminary Layout Scheme", in which specific actions and interventions were outlined in line with the strategies and the goals previously defined.

2.2.4. Phase 3 | Project

Even the third phase of the "Project", as the first of the "Analysis and critical evaluation of the territory" was divided into two consequential steps:

1. Masterplan | The first step of this phase concerns the representation of the project design proposal for the Preliminary Layout Scheme, i.e., a schematization of site-specific actions related to strategies and goals outlined in the previous "Planning" phase.

2. Toolkit of actions | In the second step, all the design solutions were analyzed in detail in a toolkit [34] aimed at clarifying their effects in environmental, social and economic terms. Moreover, the following characteristics were explained for each action: the scale of intervention, the effect produced in terms of improvements in urban quality and the actors involved in the implementation.

The third phase ended with the public presentation of the results of each group of students.

## 3. Case Study

The design workshop focuses on the Flaminio district, in particular on the Olympic Village and its relationship with the Tiber River. Rome's riversides are the expression of the millennial relationship between the river and the city made up of those elements which testify to the evolution of the urban growth [35]. Among these, the bridges, the buildings and the mobility network that follows the course of the Tiber represent the physical, functional and perceptive relationships between the natural and the urban environment [36].

The Flaminio district is located to the north of Rome, outside the Aurelian walls and remained, until the end of the 19th century, in an "isolated" dimension, identifying itself as a piece of the Roman countryside dominated by the consular road Via Flaminia, from which several alleys led to the numerous vineyards and villas of the Roman nobility. Therefore, the modern development of the district, according to the Master Plans of Rome (dated 1909, 1931, 1962 and 2008), is strictly connected with the construction of new physical connections

between the two banks of the Tiber. In this sense, the development of this area follows the number of bridges built to unify this isolated part of the city with its surroundings (Figure 2). The Sanjust Plan of 1909 conceived the continuation of the left bank of the Tiber from Ponte Milvio, one of the oldest bridges in Rome, dated 207 b.C., also called "Ponte Mollo" (Soft Bridge) for its elasticity during the numerous Tiber floods that have plagued the city of Rome from antiquity to the present [37].

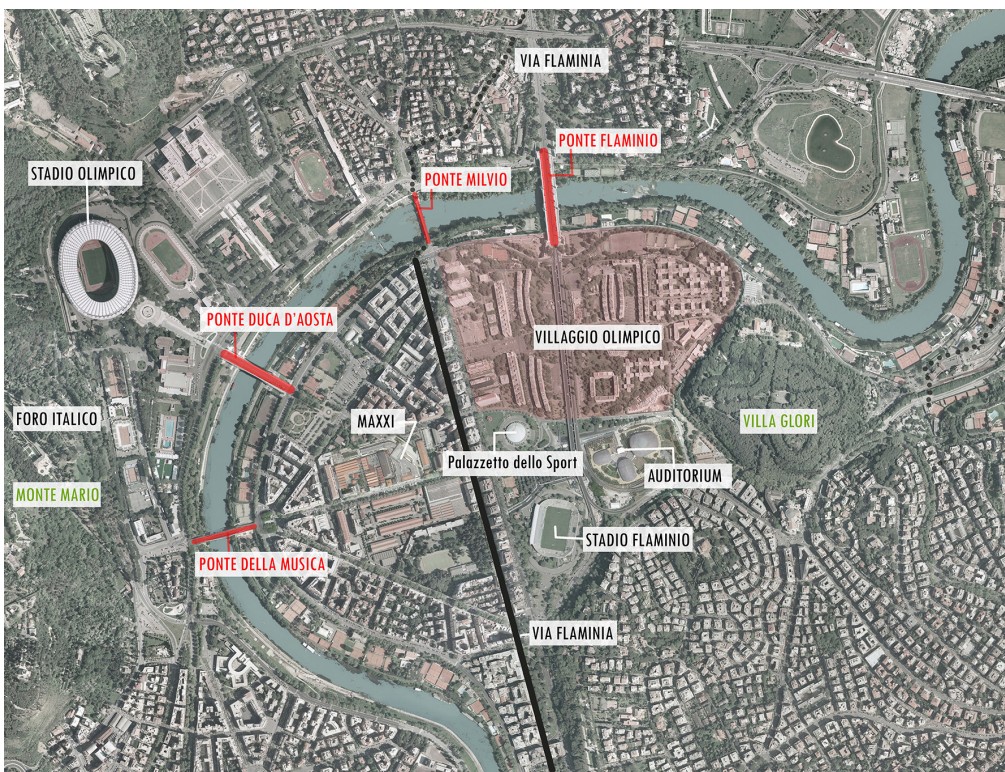

**Figure 2.** The area of the Flaminio District within evidence of the Olympic Village and the main urban references. Source: authors' elaboration.

The Plan also envisaged the trident radiating out towards the Via Flaminia that nowadays constitutes the distinctive footprint of the district. With the Universal Exhibition of 1911, the Ponte Risorgimento was inaugurated, celebrating the 50th anniversary of Rome's proclamation as the Capital of Italy. In 1916, with the proposal to transform the Via Flaminia into a great monumental promenade, including Ponte Milvio and Via Cassia, the construction of the Stadio Nazionale (later Stadio Torino, today Stadio Flaminio) and the hippodrome (Ippodromo Flaminio at Villa Glori) in the bend of the Tiber, characterized the vocation of the area for sports and recreational activities, confirmed, over the years, by the number of sports clubs and some sports facilities placed on both the banks of the Tiber, which still exist.

The Plan of 1931 introduced some changes to the urban design, strengthening the trident and providing a double connection between the Flaminio and the opposite bank of the Tiber. Two bridges, Ponte Flaminio and Ponte Duca d'Aosta, intended to link the Flaminio district with the Foro Mussolini and then Foro Italico, conceived in the late 1920s as a large sports and educational complex, a true "city of sport" whose construction started in 1928 by the Opera Nazionale Balilla just next to Ponte Milvio.

In 1935, Italy proposed Rome as the venue for the 1940 Olympic Games and, after being assigned to Tokyo, proposed it for those of 1944, which did not take place due to the start of the Second World War. The 1960 Olympic Games therefore represented a watershed in the history of the capital, in terms of urban planning. New sports facilities, new infrastructures and new parts of the city were rapidly realized, somehow out of a real urban design

and a common idea of the future of the city [38]. The Olympic Village was realized as a neighborhood of 1348 apartments, built on the site of the Campo Parioli barracks and was designed by the architects Adalberto Libera, Luigi Moretti, Vincenzo Monaco, Amedeo Luccichenti and Vittorio Cafiero, still representing a rigorous application of the urban planning theories of the modern movement. It was commissioned by INCIS, the National Institute for the Housing of State Employees (National Institute for the Housing of State Employees), which decided to build a district to host Olympic athletes, to be transformed into residences for state employees after the event [39,40].

Thirty years after the Olympics, the 1990 World Cup was decisive for district development, with two structural initiatives that contributed to its transformation: the landscaping of Piazza Mancini and the construction of the tram line between piazza Mancini and Piazzale Flaminio along via Flaminia. In 1992 the area of the Olympic Village was chosen to host the new Auditorium, with a project by Renzo Piano, and in 2002, the halls were inaugurated. In 1997, construction of the new Centre for Contemporary Arts (MAXXI) was constructed with an international competition won by Zaha Hadid (February 1999), which was inaugurated in 2011. In the same year, the new Ponte della Musica, was opened as a structuring axis between Villa Glori and Monte Mario. This new bridge completed the connections of the Flaminio district with the Foro Italico.

With the General Plan of Rome approved in 2008, the Flaminio district became part of the "Historic City", being recognized based on the historical, architectural and cultural value of a neighborhood dense with architectural works and rich in natural emergencies. Blue and green infrastructures, public spaces and cultural and sporting activities became the new keywords for the regeneration of the Flaminio district's identity.

The area's strategic role is to enhance the traces of the urban morphology, strengthening the great cultural functions of entertainment, exhibition, tourism and leisure and also defining a new sustainable mobility network.

Therefore, the choice of this study area was motivated by four main reasons:

1.  Strategic location of the Flaminio district: its proximity to the Tiber river is a fundamental component of the theme of the entire CIVIS project. From an eco-systemic perspective, the Tiber is a primary structural element both at an urban and local level. Its morphological conformation underlines the strategic relation of Rome with the specific nature of the river. Thus, the lessons learnt during the webinars concerning the case studies presented by the universities participating in the project were addressed.
2.  Criticality of the environmental, infrastructural and settlement systems that act about the proximity of the river. The river landscape belongs to the historical and environmental relationship with the city and represents a link between natural and cultural components, which assume a key role at the identity and landscape level [41].
3.  The historical–architectural value of the Olympic Village and the connections with the urban context in which it is located. Therefore, from this perspective, it is possible to easily identify the river in the urban built area of Rome, as a physical and symbolic stratification process of distinctive landscapes [42].
4.  The presence of semi-public and public spaces in the district, not yet completed, are a great potential for urban regeneration actions by defining the river as a hospitable, attractive and safe place, to reconcile landscape and perceptive values with the contemporary needs of the city and its citizens, by recognizing the river as a common good, an inclusive and sustainable infrastructure, an essential urban component and a place of aggregation, representative of the heterogeneous fluvial and urban identities [43]. In this sense, the river represents a landmark in the territory that fosters morphological, perceptive and cultural relations between public space and urban activities (Figure 3).

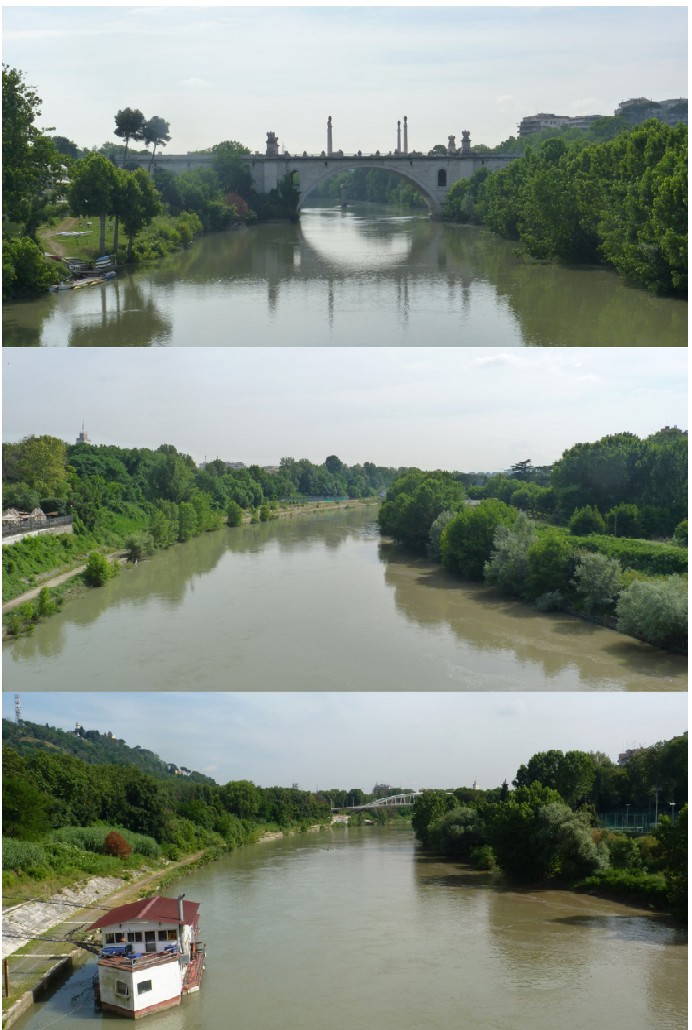

**Figure 3.** Tiber riverbanks from Ponte Flaminio (**top photo**) to Ponte della Musica (**bottom photo**). Source: photos by the authors.

## 4. Results

The design workshop activities produced different results for each group that carried out the work independently, supported by teachers and tutors through specific contributions and the monitoring of daily progress. Below are the results obtained, dealing with Phase 1: Analysis and critical evaluation of the territory; Phase 2: Planning; and Phase 3: Project at different scales, using supporting materials, such as drawings, 3D models and references provided by teachers and tutors. Then, the definition of the Toolkit of Actions has been illustrated to show the general master plan with the specific goals and actions elaborated on the sensitivities of each student involved, highlighting the peculiarities of the project area according to different points of view.

### 4.1. Phase 1: Analysis and Critical Evaluation of the Territory

The groups promoted the reading of the three systems (Environmental System, Settlement System, Infrastructural System) in a joint way to highlight the relationships among Resources, Criticalities and Potential, illustrating the priority areas of intervention, through the analysis.

### 4.2. Resources

Accessibility and safety represent the main aspects to address. Moreover, the complete lack of a relationship between the neighborhood and the river that represents more of a

barrier than a connection is another criticality. At the same time, the clear street hierarchy can represent a strength for accessibility within the district, providing both local distribution and urban connectivity. Building archetypes of the Olympic Village are representative of the Modern Architecture Movement being valuable elements conceived through the integration of private and public open spaces, defining a proper building scale for the human dimension. The other buildings, with sporting and cultural vocation, represent landmarks in the urban context and therefore a great territorial resource. The building layout and the presence of large green areas in the surroundings (Monte Mario, Villa Glori) ensure a good level of microclimatic comfort during the year, constituting one of the peculiarities of the district, offering large outdoor public spaces with commercial spots and spaces for events for the inhabitants (Figure 4).

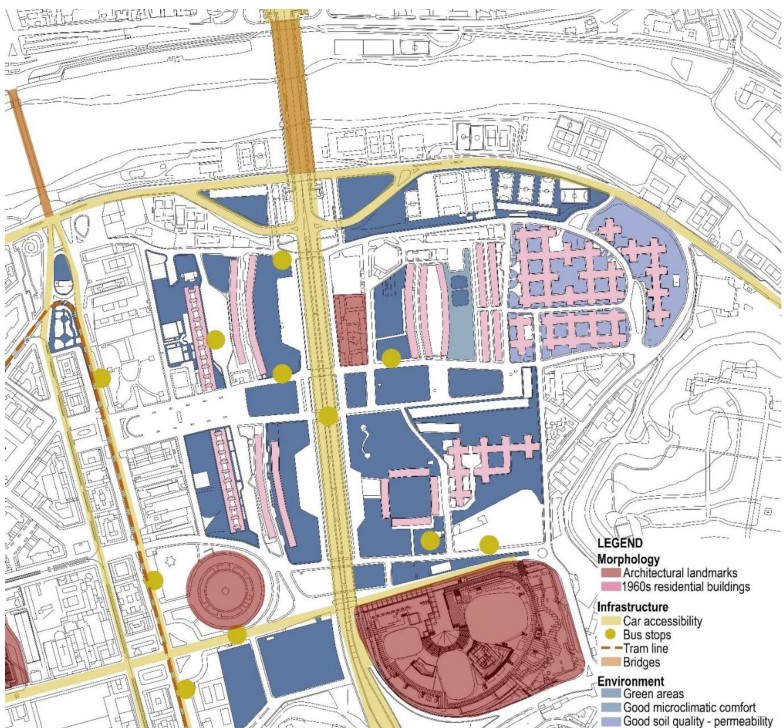

**Figure 4.** Phase 1: Analysis and critical evaluation of the territory—Resources of the Olympic Village. Source: Elaboration of Working Group 1: Ligia Gherman (SUR), Evelien Van den Bruel (ULB), Mahtab Seyedabadi (SUR).

### 4.3. Criticalities

Some critical issues have been determined, such as the inefficiency of public transport and the lack of road maintenance. This compromises the accessibility and safety of the neighborhood, making it difficult to navigate, with corresponding social consequences. Rising car traffic in the Flaminio district and mainly in the Olympic Village area leads to poor air quality due to high levels of pollution, weighing on the comfort of open spaces, affected by heat island phenomena in the summer period. In addition, the presence of the Viaduct of Corso Francia determines a social criticality, threatening the safety and comfort conditions of the open spaces

The existing vegetation presents a general level of carelessness, caused by the absence of suitable maintenance plans, showing notable criticalities of the ecological system. The large open spaces and squares are not widely used by residents and people who visit the neighborhood, where only a few commercial activities take place (Figure 5). Finally, the river, due to several issues related to the flooding risk, soil sealing and pollution, represents a criticality for the hydrographic network.

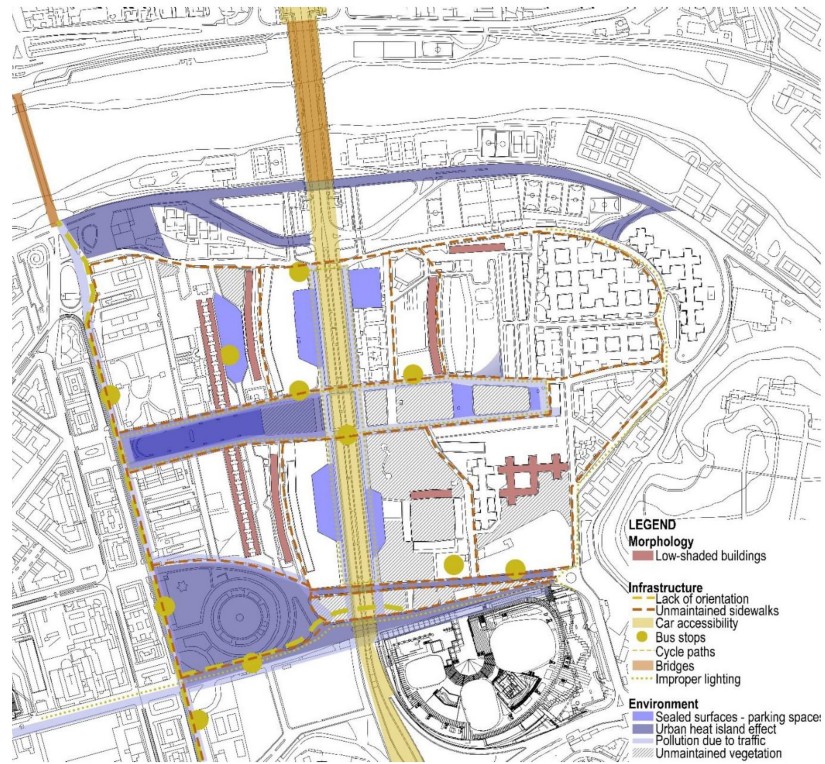

**Figure 5.** Phase 1: Analysis and critical evaluation of the territory—Criticalities of the Olympic Village. Source: Elaboration of Working Group 1: Ligia Gherman (SUR), Evelien Van den Bruel (ULB), Mahtab Seyedabadi (SUR).

*4.4. Potentials*

The Tiber river represents the main potential of the area, providing high levels of biodiversity, guaranteed by the presence of many plant and animal species. Further, the connection between Villa Glori and the Monte Mario Natural Reserve represents a potential for the construction of the Ecological Network.

The large road sections and the presence of smaller roads define a hierarchy for the local mobility, representing an opportunity from a social point of view. The open spaces and large vegetated surfaces, characterized by rows of trees and existing ecological networks, constitute a precious resource for rainwater management and avoiding the summer heat island phenomena, representing a development potential for the Olympic Village (Figure 6).

Phase 2: Planning

The analysis of resources, criticalities and potentials led to the identification of strategies for each system, in order to do the following:

- Enhance the architectural identity of the Olympic Village and the relationship between buildings and open spaces;
- Encourage relationships between residents and visitors (Morphological System), define a hierarchy for mobility and ensuring accessibility to the Olympic Village (Infrastructural System) and improve summer comfort conditions (Environmental System).

Phase 3: Project

In the Project phase, the groups have defined different solutions in order to respond to the planning strategies outlined in the previous phase (Planning), declining the objectives according to specific regenerative actions. All the design solutions were then analyzed in detail in the Toolkit of Action aimed at clarifying their effects in environmental, social and economic terms. Each action is explained in order to give information about the scale of intervention, the effect produced in terms of the improvement of urban quality and possible stakeholders involved in the implementation.

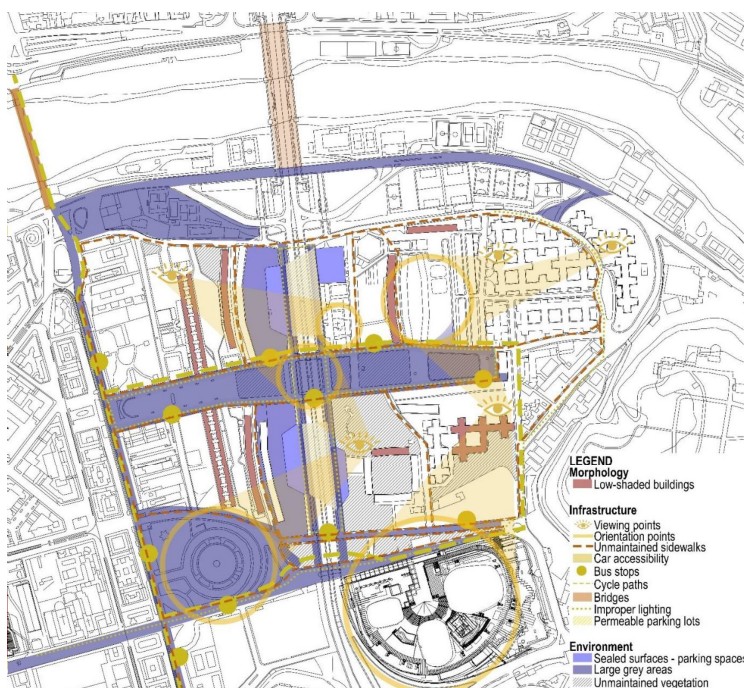

**Figure 6.** Phase 1: Analysis and critical evaluation of the territory—Potential of the Olympic Village. Source: Elaboration of Working Group 1: Ligia Gherman (SUR), Evelien Van den Bruel (ULB), Mahtab Seyedabadi (SUR).

### 4.5. Masterplan 1

In this Masterplan, the attention focuses on the central area of the Olympic Village and integrates solutions to implement permeable surfaces, such as draining pavements and vegetation, designing new pedestrian paths and redefining car traffic through a clear hierarchy of distances.

Modelling the soil and unsealing the surfaces have been set as the main strategies of the project, introducing new vegetated spaces. Actions aimed at sustainable rainwater management have been defined, through a circular approach, for the collection, treatment and reuse of water resources, providing a low cost of maintenance of open spaces, mitigating the urban heat island effects and reducing the runoff risk affecting the project area (Figure 7).

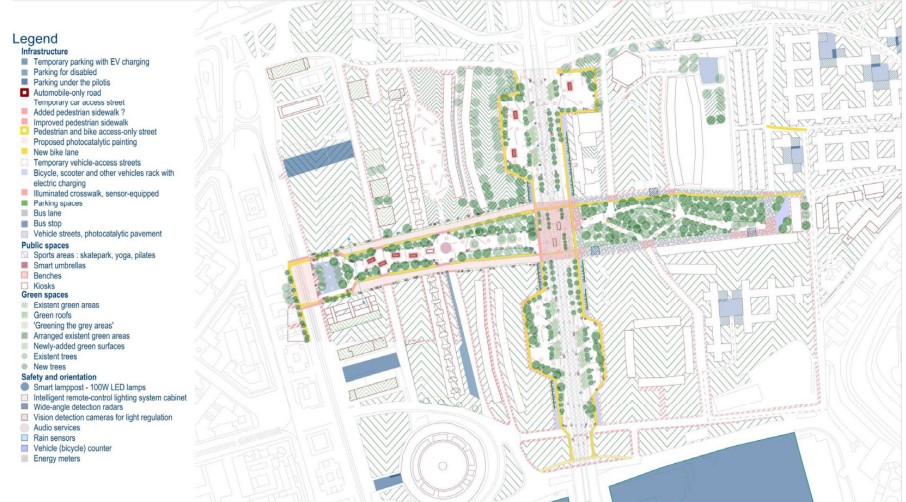

**Figure 7.** Phase 3: Project Masterplan. Source: Elaboration of Working Group 1: Ligia Gherman (SUR), Evelien Van den Bruel (ULB), Mahtab Seyedabadi (SUR).

### 4.6. Toolkit of Actions

The main objectives and strategies were articulated and organized with the related actions indicated for each system in the toolkit. The specific goals and actions of the Olympic Village have taken into greater detail the central area constituted by the progression of large open spaces designed to connect the neighborhood and improve its livability (Figures 8 and 9).

| CHALLENGE | GOALS | STRATEGIES | ACTIONS |
|---|---|---|---|
| Undefined spaces: Safety, communication, economy. | Mixed use according to time. Redefine and strengthen identity of area. | Increase the communication between the two part of neighborhood | Parking lots. Green gardens as Common activities for different ages and genders. |

| CHALLENGE | GOALS | STRATEGIES | ACTIONS |
|---|---|---|---|
| Uncomfortable spaces: Heat island, poor water management, low impermeability. | Improve water management. Reduce heat island under bridge. Control noise pollution | Increase evapotranspiration. | Ecological Park(ing) site. Veracity of trees and vegetation. Capture, filter and reuse of runoff water for agriculture and selling them on market. |

| CHALLENGE | GOALS | STRATEGIES | ACTIONS |
|---|---|---|---|
| inaccessible spaces: Safety, mobility. | Redefine parking lots. Arrange accessibility for cars and pedestrians. | Use of different paving types. | Change the asphalt areas into natural terrains that reintroduce biodiversity. |

**Figure 8.** Toolkit of Actions for the vertical axis of the central public spaces in the Olympic Village. Source: Elaboration of Working Group 1: Ligia Gherman (SUR), Evelien Van den Bruel (ULB), Mahtab Seyedabadi (SUR).

| CHALLENGE | GOALS | STRATEGIES | ACTIONS |
|---|---|---|---|
| Undefined spaces: Safety, communication, economy. | Increase identity and social life. Boost of economy. | Connection of residential blocks with central axes. Make axes safe and useful for everybody. | Creation of encounter places for temporary events. Kiosks on the critical connection points with following up activities. Add verity of urban furniture. |

| CHALLENGE | GOALS | STRATEGIES | ACTIONS |
|---|---|---|---|
| Uncomfortable spaces: Heat island, poor water management, low impermeability. | Reduce heat islands under the bridge and empty areas. | Reduce empty areas. Increase plants diversity. | Reduce asphalt amount. Add canopies and smart umbrella. Rainwater management: Capture, storage, reuse and infiltration. |

| CHALLENGE | GOALS | STRATEGIES | ACTIONS |
|---|---|---|---|
| inaccessible spaces: Safety, mobility. | Safety for pedestrian. | Highlight crossing roads and prioritize pedestrians. Visibility of crossing roads | Smart lighting. Increasing pavement height. Road marks, traffic boards flashlights |

**Figure 9.** Toolkit of Actions for the horizontal axis of the central public spaces in the Olympic Village. Source: Elaboration of Working Group 1: Ligia Gherman (SUR), Evelien Van den Bruel (ULB), Mahtab Seyedabadi (SUR).

### 4.7. Masterplan 2

In this Second Masterplan, two new urban parks along the Tiber river banks have been designed to mitigate the flooding risk providing the ecosystem benefits of the vegetation and the relationship between the city and the river, generating opportunities for socio-ecological connections (Figure 10).

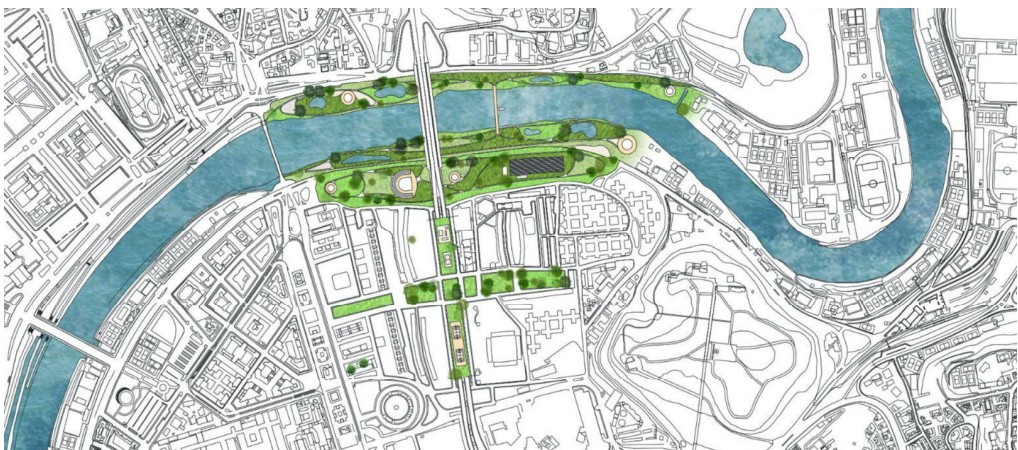

**Figure 10.** Project Masterplan. Source: Elaboration of Working Group 2: Augustina Lezcano (UAM), Irene Maroncelli (SUR), Agahi Najafabadi Parnian (SUR), Khawla Rachadi (SUR).

The pictures show the relationships between the two urban parks, in which the functions and integrated solutions adopted are designed, where the contribution of ecological-environmental benefits is provided by the presence of vegetation that contributes to increasing the air and soil quality in an inclusive public space (Figure 11).

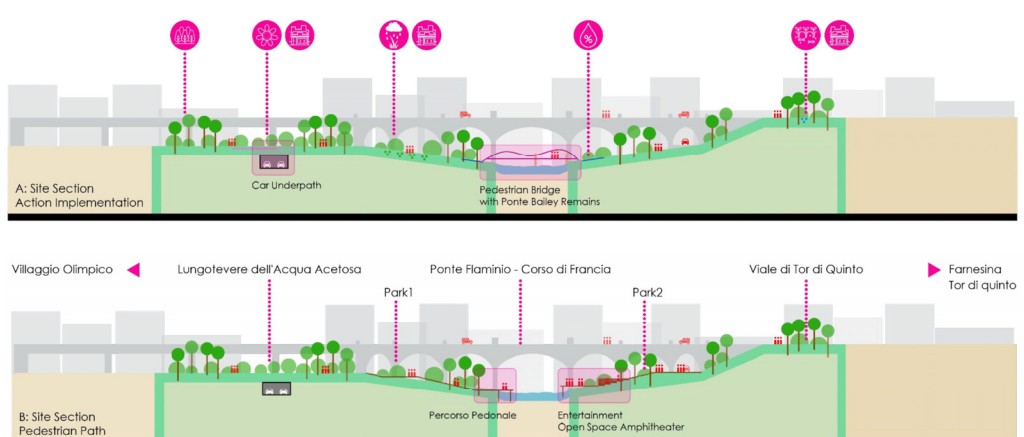

**Figure 11.** Sections on the twin parks along the Tiber river. Functional areas. Source: Elaboration of Working Group 2: Augustina Lezcano (UAM), Irene Maroncelli (SUR), Agahi Najafabadi Parnian (SUR), Khawla Rachadi (SUR).

### 4.8. Toolkit of Actions

The "Twin Parks" project defined interventions to create new ecological and infrastructural connections, through an inclusive and attractive public space, able to mitigate the risks highlighted in the analysis phase, implementing solutions to promote external comfort and reducing the urban heat island phenomena (Figure 12).

### 4.9. Masterplan 3

In this case, the project's concept is the "Blooming Village" according to the principles of urban circular economy, to foster the functionalization of public spaces and buildings. To provide a safe and accessible neighborhood, an efficient infrastructural system through the hierarchy of vehicular traffic and cycle–pedestrian paths has been designed, incentivizing the modal shift and implementing strategies and solutions aimed at the sustainable management of water resources.

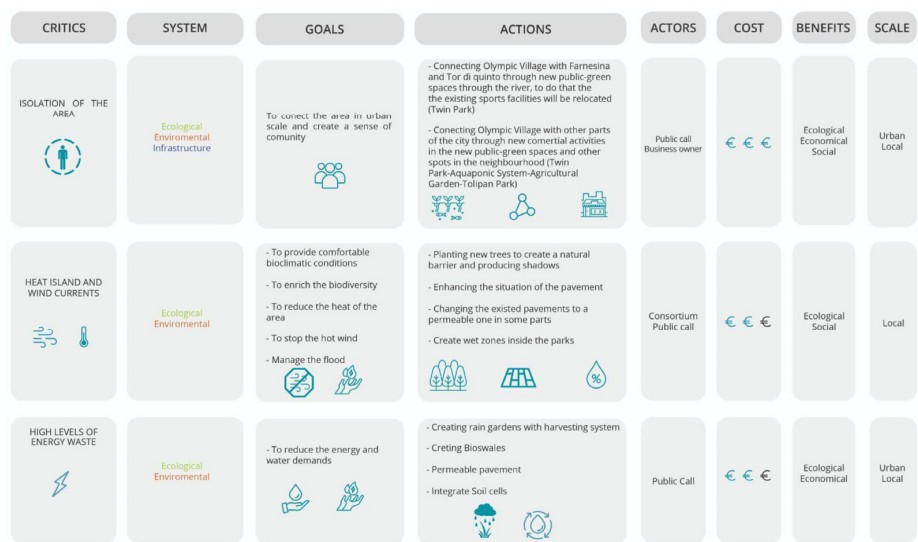

**Figure 12.** Toolkit. Source: Elaboration of Working Group 2: Augustina Lezcano (UAM), Irene Maroncelli (SUR), Agahi Najafabadi Parnian (SUR), Khawla Rachadi (SUR).

Through a bottom-up approach, the involvement of the residents of the Olympic Village is ensured, encouraging virtuous processes for the transformation of the spaces and increasing the attractive power of the neighborhood. Carrying out collective activities allows for the development of a sense of community, creating new job opportunities and, at the same time, reducing the maintenance costs of public open spaces for a resilient community (Figure 13).

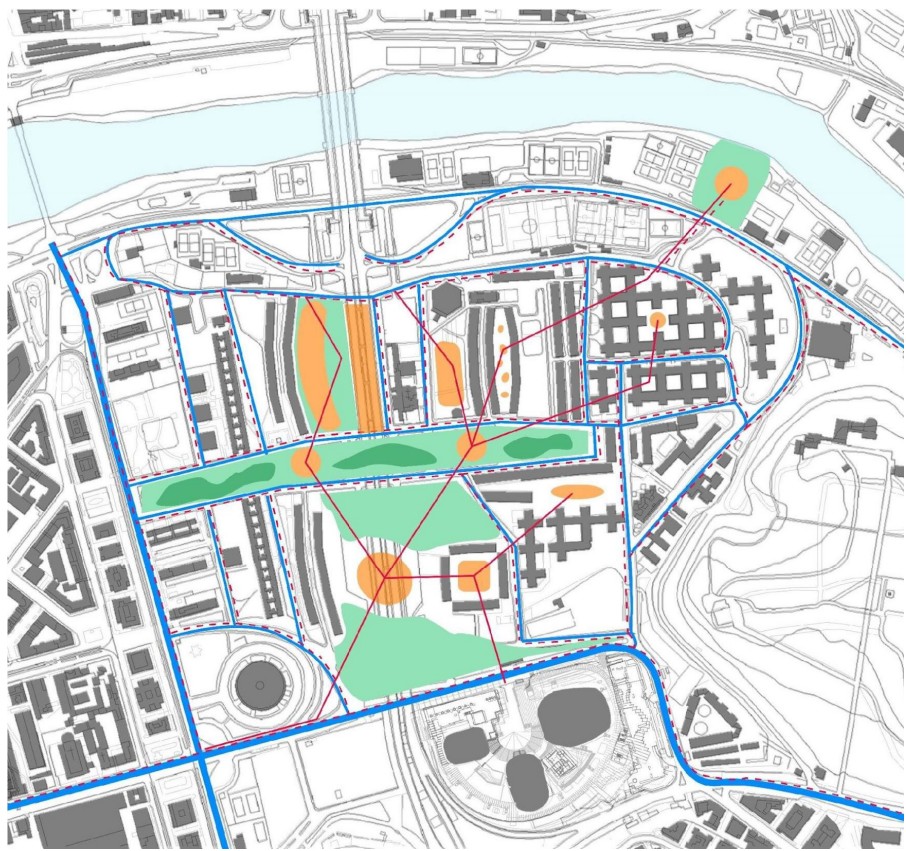

**Figure 13.** Project Strategies. Source: Elaboration of Working Group 3: Luca Annicchiarico (SUR), Carlotta Fabrizio (SUR), Angèle Khoury (ULB).

### 4.10. Toolkit of Actions

For each strategy represented in the general Masterplan, different actions are shown, as indicated in the toolkit, listing the different solutions for each system, illustrating how each action is linked to the specific project goal, as illustrated in the following map and figure (Figures 14 and 15).

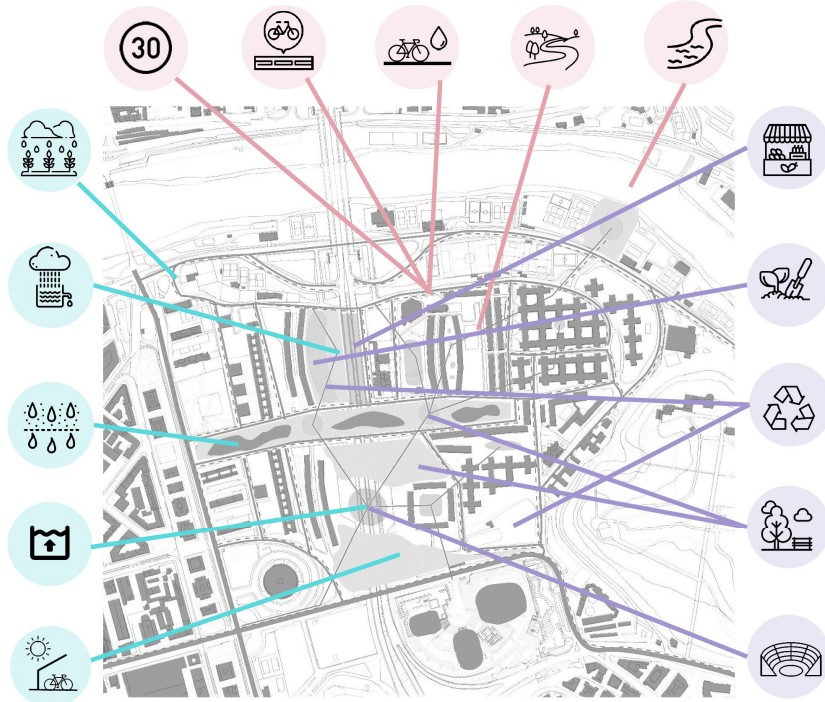

**Figure 14.** Toolkit of Actions with specific actions. Source: Elaboration of Working Group 3: Luca Annicchiarico (SUR), Carlotta Fabrizio (SUR), Angèle Khoury (ULB).

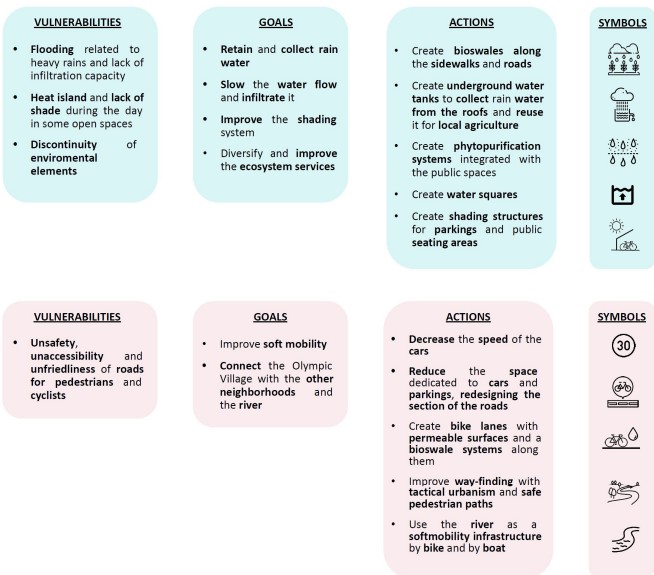

**Figure 15.** *Cont.*

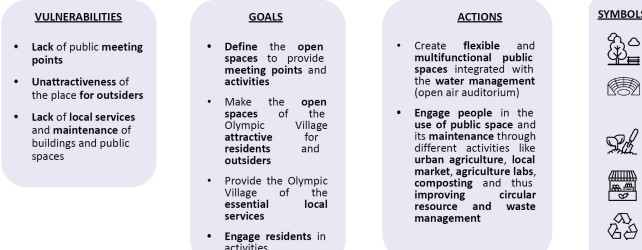

**Figure 15.** Toolkit of Actions for the Environmental, mobility and Public Space systems. Source: Elaboration of Working Group 3: Luca Annicchiarico (SUR), Carlotta Fabrizio (SUR), Angèle Khoury (ULB).

## 5. Conclusions

The in-depth knowledge of territorial, cultural and environmental heritage and the awareness of risks and vulnerabilities can lead together to the mitigation of the effects of disasters by focusing on adaptive behaviors instead of reactive ones, all this to the end of improving the resilience of local communities [44]. The capability to define and network innovative place-based and nature-based solutions fosters flexible and incremental territorial actions as the fundamental components of urban regeneration strategies, concerning long-standing sustainable development goals, as well as concerning pandemic emergencies [45]. Therefore, the innovation of public space is one of the most relevant drivers of the paradigm shift toward resilient, inclusive and green cities, strengthening ecological connections, as well as enhancing social and cultural networks [46].

In this framework, the environmental regenerative design strategies proposed within the RivEr/Generation_LAB activities on the Olympic Village of Rome have been considered from a twofold point of view, in order to highlight the strong integration existing, at all scales, between urban design experimentation and the environmental sustainability of the interventions.

On a local scale, in order to minimize times and costs of construction and reduce its environmental impact, the regenerative scenario levers on natural materials, such as wood and plants, used both in flexible and modular systems placed in the open space.

The buildings' roofs represent another "layer" to be re-designed to mitigate the UHI effect, using reflecting materials (i.e., with high albedo values) and green and brown roofs to mitigate the microclimate. New tree-lined rows define the routes and vegetated linear barriers, which mitigate traffic noise and air pollution while improving the thermo-hygrometric conditions through evapotranspiration and shading.

New permeable surfaces—on ground (rain gardens and bio-swales) and on the buildings' roof (extensive and intensive green roofs) —contribute to the reduction of the UHI effect, in summer. Increased permeable surfaces and a new water square located in the south edge of the complex also allow a significant amount of rainwater to be collected, ensuring sustainable water management.

On a larger scale, the restoration of the socio-ecological connection with the Tiber River and other relevant green areas, such as Villa Glori and Monte Mario, is proposed. Focusing on tree planting design and new pedestrian and cycle paths, the design aims at guaranteeing improved connectivity among the residential areas, the Tiber River, and the neighboring areas, structuring new public and semi-public areas, for the benefit of people and the environment.

Moreover, the development of the above-mentioned strategies and actions assumes even more strength if considered in the framework of the innovative planning approach of eco-districts to climate-proof urban design, which articulate adaptation and mitigation planning and design solutions with particular attention to soil, vegetation, water and energy and integrate nature-based solutions with the new issues of smart and inclusive cities [47,48].

Finally, the pivotal role given to the relationship between the Tiber river and the Olympic Village strengthens the aim to enhance water landscapes and cultural heritage,

together with public and private activities, for a safer and more accessible fruition of natural and urban spaces, which concretely contribute to transform rivers into new urban centralities, "common goods" capable of linking resilience with inclusiveness in the urban-built environments of contemporary cities.

A new territorial articulation in which ecology restores to the landscape its performative function with the aim to be the new infrastructure able to support future transformation and development activities [49,50].

**Author Contributions:** Conceptualization, C.M.; methodology, C.M. and F.R.; software, C.M. and F.R.; validation, C.M. and F.R.; format analysis, C.M. and F.R.; investigation, C.M. and F.R.; resources, C.M.; data curation, C.M. and F.R.; writing-original draft preparation, C.M. and F.R.; writing-review and editing, C.M. and F.R.; supervision, C.M.; project administration, C.M.; funding acquisition, C.M. All authors have read and agreed to the published version of the manuscript.

**Funding:** This research was financed by the CIVIS Steering committee CIVIS Hub's 'Hub4 Cities, Territories & Mobilities' Call for proposals 2021, Scientific Coordinator Prof. Carmela Mariano, Department PDTA, Sapienza University of Rome Identification number 612648-EPP-1-2019-1-FR-EPPKA2-EUR-UNIV.

**Institutional Review Board Statement:** Not applicable.

**Informed Consent Statement:** Not applicable.

**Data Availability Statement:** Not applicable.

**Acknowledgments:** The paper is the result of a shared reflection by the authors. However, paragraphs 1 and 2 are to be attributed to Carmela Mariano; paragraphs 3 and 4 are to be attributed to Francesca Rossi; and paragraph 5 is to be attributed to both authors.

**Conflicts of Interest:** The authors declare no conflict of interest.

**Appendix A**

The design Workshop "RivEr/Generation_LAB. Linking resilience with inclusiveness in the urban built environment of Rome, Brussels, Madrid", open to master's degree students from universities in the CIVIS Network, was held from 4 to 9 July 2022 at the PDTA Department, Sapienza University of Rome. The activities related to the Workshop were coordinated by the PDTA Department of Sapienza, University of Rome—SUR (Principal Investigator: Carmela Mariano with Maria Beatrice Andreucci and Francesca Rossi)—with the collaboration of the Université libre de Bruxelles—ULB (Referent professor: Didier Vancutsem, Master's degree Landscape Architecture and Urban Planning)—and the Universidad Autónoma de Madrid—UAM (Referent professors: Maria del Mar Alonso Almeida; Beatriz Narbona Reina and Fernando Borrajo Millan; Maria Escat).

The cycle of six webinars focused on specific subject areas.
In particular:

- The first webinar, "Urban Planning for RivEr/Generation" (12 April 2022), was coordinated by the Sapienza University of Rome, SUR.

The webinar, introduced by Francesca Rossi (SUR) and with the closing remarks of Maria Beatrice Andreucci (SUR) and Carmen Mariano (SUR), hosted three specific contributions: Historical plans and urban fabric evolution in the Flaminio area by Bruno Monardo, PDTA Department, Sapienza; Rome and the Tiber Designing in flood risk areas, framing the topic first in the Roman context, supporting it with international case studies that address the phenomena linked to climate change in the urban context with design solutions by Alessandra de Cesaris, PDTA Department, Sapienza) and Water landscapes: Strategies for adapting cities to the effects of climate change and the three macro-strategies for a Resilient urban regeneration: defense, adaptation and relocation, supported by the presentation of case studies, by Marsia Marino, PDTA Department, Sapienza.

- The second webinar, "Mobility. The case of Madrid" (28 April 2022), was coordinated by the Universidad Autónoma de Madrid, UAM.

The webinar, by Beatriz Narbona (UAM), focused on the topic of mobility, providing a practical overview of mobility trends, with a multidisciplinary approach from social, societal and economic perspectives, always considering the involvement of citizens. The topic of mobility was addressed starting with an overview of smart cities from the sustainable mobility point of view, introducing the topics of Smart Mobility Plans and the MaaS (Mobility as a Service) concept, the integration of various transport services into a single mobility service accessible on demand and with a single payment, and then focusing on the case study of Madrid, in particular by presenting the contents of the project "Madrid 360 Sustainable Mobility: A cleaner air for everyone".

- The third webinar "Urban Economy" (12 May 2022), was coordinated by the Universidad Autónoma de Madrid, UAM.

The webinar, by Maria Escat (UAM), focused on urban economic issues, thus promoting interdisciplinarity and allowing students to experiment with the main objective and value of the entire initiative, which takes advantage of the specific expertise of the three universities. In this case, the meaning of "Urban Economics" applied to the case study of Madrid was explored, supported by a general presentation of the evolution of the city in accordance with the growth phases of the Manzanares River, citing the 'Madrid Rìo' project and making economic considerations based on the obtained results.

- The fourth webinar "Ecology" (31 May 2022), was coordinated by Université Libre de Bruxelles, ULB.

The lecture explored two core themes of the project: the phenomenon of flooding involving the Veneto Region, with a contribution entitled: Veneto 2100- Living with Water by Marco Ranzato (Roma Tre University) and Ecological Challenges and Nature Based Solutions by Didier Vancutsem (ULB)

- The fifth webinar "Landscape Architecture" (9 June 2022), was coordinated by the Université Libre de Bruxelles, ULB.

The webinar, entitled "Navigating in Brussels waters" was introduced by Didier Vancutsem and followed by two in-depth presentations related to two different approaches to the topic: a first engineering approach, with the contribution entitled "Dialog of rainwater in the public space" by Dimitri Crespin (Vrije Universiteit Brussel), and a second Architectural Landscaping approach, with the contribution entitled "Design private spaces to engage citizens" by Andrea Aragone (ULB).

- The sixth webinar "Environmental Design" (4 July 2022) was coordinated by the PDTA Department of Sapienza University of Rome.

The webinar, by Alessandro Stracqualursi (SUR), focused on the topic of water: risks and opportunities, consistent with the sixth goal "Clean water and sanitation", belonging to the 17 Sustainable Development Goals. The topics of the natural water cycle, urban water cycle, water categories, integrated water cycle management and water uses on the urban scale of use were then explored. Finally, the theme of water pertaining to climate change was addressed, specifically the risks arising from scarcity, sea level rise and flooding, providing case studies of adaptation solutions in urban contexts and rural areas.

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
