# Peer review of "RivEr/Generation_LAB-Linking Resilience with Inclusiveness in the Urban-Built Environment of Rome"

_sustainability, doi:10.3390/su15064774_

Round 1

Reviewer 1 Report

This is an article to disseminate the activities and results obtained from the "RivEr/Generation_LAB" project, which proposes an urban intervention methodology with the aim of establishing new relationships between cities and their natural environment.

The article presents numerous textual inconsistencies, which make it difficult to understand. For example, in line 199, it is written "The workshop 19], open to master's degree students from universities in the CIVIS". What exactly is this "Workshop 19] ..."?

Some sections of the article do not make much sense in the context of a dissemination article like this one. For example, section "2. Materials and method" does not make sense to call it "materials", since the alleged research did not involve "materials" per se. Nor do I think it makes sense to name in the article the members of the working group in charge of developing the "RivEr/Generation_LAB" project.

The title of the map in Figure 4 is not understood, or the relationship between the image and the text "Analysis and critical evaluation of the territory-Resources" is not understood. Do the authors believe that this map really represents a "critical evaluation of the territory"? Moreover, it does not seem that the maps or drawings in Figures 4, 5 and 6, represent exactly the same territory of the Flaminio District as depicted in Figure 2. Even worse, the maps are not even to scale.

In the opinion of this reviewer, text and figures are somewhat chaotically and poorly distributed throughout the article. Furthermore, it seems that the authors do not follow the editing rules of the template provided by the journal "Sustainability".

Reviewer 2 Report

The paper “RivEr/Generation_LAB. Linking resilience with inclusiveness in the urban built environment of Rome” provides an informative guide to a fascinating project. It presents the results of an inter-campus collaborative project of universities in Rome, Bruxelles, and Madrid. This collective developed a methodology that integrates the interests of seemingly competing systems in urban environments (ecology, mobility, economy) with sustainability goals. The collaborative, interdisciplinary, and international project proposes a methodology of intervention in a specific district of Rome, Italy, that links resilience with inclusiveness. Invigorating and renewing an urban area along the Tiber River to increase and improve relationships between the built urban landscape, green urban spaces, and communities of users. Sustainable Development Goals, the Millenium Ecosystem Assessment, and the New European Bauhaus provide guiding principles and frameworks. The co-authors address the need for sustainable management of natural and urban resources with a commitment to ethical and equitable development and to building inclusive communities. Their paper offers a detailed account of a project designed to improve microclimatic conditions in urban areas and the quality of life in those areas through intentional design principles of ecological regeneration. Economic, cultural, and social resilience increase through this kind of urban redesign. The methodological framework for this project includes pedagogical components, e.g. a series  of webinars and on-site workshops where diverse approaches are integrated to promote the regeneration of urban riverscapes. The case study introduced in detail is on the Tiber River in the Olympic Village area in Rome in the Flaminio district. Three working groups provide readers with detailed accounts of their analyses, including detailed illustrations. The conclusion offers insights into how the restoration of the socio-ecological connection of urban spaces and the Tiber River is beneficial to people and the environment and contributes to building resilient communities.  The strength of this paper resides in its capacity to use a case study of a specific space to demonstrate replicable processes of  analysis (including a toolkit) that integrate perspectives from  different disciplines and stakeholders.  scientific content of the manuscript and should be specific enough for the authors to be able to respond.

Specific comments : The paper is well structured and engages readers with an effective mix of contextual information and concise accounts of procedures, challenges, and solutions addressed in webinars and workshops that lead to the well founded recommendations of improvements for a specific case study of an urban area in need of improvements, namely  the Olympic Village area in Rome along the Tiber River. The  cited references are relevant and recent and the design of the case study is clearly articulated with goals, guiding principles, tool kits, analysis, and outcome summaries. The illustrations are an integral part of the paper. The  figures/tables/images/ are appropriate and the  conclusions are consistent with the evidence and arguments presented. 

Reviewer 3 Report

Review comments on the manuscript (sustainability-2130378) titled by RivEr/Generation_LAB. Linking resilience with inclusiveness in 2 the urban built environment of Rome.

It is an interesting manuscript that provide a methodology and major results of RivEr/Generation_LAB project to solve the relationships between urban development and natural environment. The proposed regenerative scenario and toolkits levers on natural materials like wood and plants, used both in flexible and modular systems placed in the buildings’ roofs, tree-lined rows, permeable surfaces, et al. In my opinion, the structure of manuscript, especially in section Introduction and section Materials and method, should be improved to draw the picture of RivEr/Generation_LAB project clearly. What is relationship between the case study and RivEr/Generation_LAB project? You could some maps to show the area and scope of the whole project and the location of the case. If I understand right, the section Results is a part of case study and the regenerative scenario is worthy of promotion and application for other regions. Furthermore, it is a pity that the manuscript only gives the regenerative scenario based on some conceptualization and methodology but without concrete and scientific monitoring data to verify the actual impacts on regenerative scenario.

Round 2

Reviewer 1 Report

The article has improved substantially after the revision made by the authors, so in my opinion it can be published.

Author Response

Response to Reviewer 1

I would like to thank the reviewer for her/his constructive comment. However, responding to the specific note concerning if the article was adequately referenced, some bibliographic references have been added where the authors have deemed appropriate

Answer to the reviewer’s specific comments about content and formatting issues:

The article has improved substantially after the revision made by the authors, so in my opinion it can be published.

References number 7, 14, 37, 45 have been added to better specify the article’s scientific framework.

Reviewer 3 Report

The author improved the manuscript and replied most of my concerns. I suggest the author do not submit the manuscript in modified mode, which is very mess and is uncomfortable to read. Please submit the manuscript with the marked revision by using the red color. I have no more comments on the revised manuscript.

Author Response

Response to Reviewer 3

I would like to thank the reviewer for her/his constructive comments and suggestions. However, some bibliographic references have been added where the authors have deemed appropriate to better specify the article’s scientific framework (References number 7, 14, 17, 35, 36, 39, 40, 47, 50)

Answer to the reviewer’s specific comments about content and formatting issues:

1 The author improved the manuscript and replied most of my concerns. I suggest the author do not submit the manuscript in modified mode, which is very mess and is uncomfortable to read. Please submit the manuscript with the marked revision by using the red color. I have no more comments on the revised manuscript.

Sorry for the difficulty in reading but the paper with track changes is the required format by the editorial office